# Indoleamine-2,3-Dioxygenase in Thyroid Cancer Cells Suppresses Natural Killer Cell Function by Inhibiting NKG2D and NKp46 Expression via STAT Signaling Pathways

**DOI:** 10.3390/jcm8060842

**Published:** 2019-06-12

**Authors:** Arum Park, Yunjeong Yang, Yunhee Lee, Mi Sun Kim, Young-Jun Park, Haiyoung Jung, Tae-Don Kim, Hee Gu Lee, Inpyo Choi, Suk Ran Yoon

**Affiliations:** 1Immunotherapy Research Center, Korea Research Institute of Bioscience and Biotechnology, Yuseong-gu, Daejeon 34141, Korea; sjllar@kribb.re.kr (A.P.); hike14@kribb.re.kr (Y.Y.); heeya@kribb.re.kr (Y.L.); misun.kim1016@gmail.com (M.S.K.); pyj71@kribb.re.kr (Y.-J.P.); haiyoung@kribb.re.kr (H.J.); tdkim@kribb.re.kr (T.-D.K.); hglee@kribb.re.kr (H.G.L.); 2Department of Functional Genomics, University of Science & Technology, Daejeon 34113, Korea; 3Department of Pharmacology, College of Pharmacy, Chungnam National University, Daejeon 34134, Korea

**Keywords:** NK cells, thyroid cancer cells, indoleamine 2,3-dioxygenase, kynurenine, NKG2D, NKp46, STAT1, STAT 3

## Abstract

Natural killer (NK) cells are key players in the immune system. They use receptors on their cell surface to identify target cells. However, to escape being killed by the immune system, cancer cells such as thyroid cancer cells, use various methods to suppress the function of NK cells. Thus, this study aims to elucidate how thyroid cancer cells downregulate NK cell function in a co-culture system. We found that thyroid cancer cells suppress NK cell cytotoxicity and inhibit the expression of activating receptors, such as NKG2D and NKp46, by regulating indoleamine 2,3-dioxygenase (IDO). Also, thyroid cancer cells produce kynurenine using IDO, which causes NK cell dysfunction. Kynurenine enters NK cells via the aryl hydrocarbon receptor (AhR) on the surfaces of the NK cells, which decreases NK cell function and NK receptor expression via the signal transducer and activator of transcription (STAT) 1 and STAT3 pathways. In addition, STAT1 and STAT3 directly regulated the expression of NKG2D and NKp46 receptors by binding to the promoter region. Conclusively, NK cell function may be impaired in thyroid cancer patients by IDO-induced kynurenine production. This implies that IDO can be used as a target for thyroid cancer therapeutics aiming at improving NK cell function.

## 1. Introduction

NK cells inhibit tumor cells through several pathways. Targets are either recognized and directly killed by released granules or antibody-dependent cellular cytotoxicity (ADCC), or indirectly killed by other immune cells that gather due to cytokine release by the NK cells [1,2,3,4]. Activating and inhibitory receptors of NK cells play an important role in the activation of NK cells [5]. NK cells use their receptors that interact with ligands of target cells to determine the fate of these cells. When ligands bind, these activating and inhibitory receptors cooperate and decide whether to exert NK cell cytotoxicity on target cells [6]. Some important NK activating receptors include NKG2D, DNAM1, natural cytotoxic receptors (NCR) (NKp46, NKp30, and NKp44), and CD16 which are involved in antibody-dependent cytotoxicity (ADCC) [7].

Despite their ability to directly remove cancer cells, NK cells in tumor environments do not easily access the tumor site or are inhibited by other factors released by cancers [8,9]. One of such factors, indoleamine 2,3-dioxygenase (IDO), is mainly involved in T cell immunosuppression and is known to help tumor growth. Interferon (IFN)-γ has been known as an inducer of IDO mechanism [10]. IDO enables tumor cells to escape the immune system by producing kynurenine, a tryptophan metabolite that selectively impairs the growth and survival of T-cells in tumor microenvironments [11,12]. Kynurenine is also known to be involved in the functional degradation of NK cells, but its mechanism of NK cell function downregulation remains unclear. Kynurenine enters the cells via AHR (aryl hydrocarbon receptor) [13]. It interacts and regulates signal transduction using several transcription factors, including signal transducer and activator of transcription (STAT)1, STAT3, STAT5, Pai-2, Sp1, c-maf, and Bach2, in certain cell types [14,15,16,17,18].

The JAK-STAT signaling pathway plays an important role in NK cells, as most cytokines that can activate or block NK cells are known to regulate it [19]. IL-2, a cytokine that plays an important role in NK cell proliferation and receptor expression, is known to activate STAT1, 3, and 5 [20]. It is known that STAT5 is activated by IL-15, and that STAT1 and 3 are activated by IL-21, which leads to maturation, proliferation, and activation of NK cells [21,22]. In addition, various types of STAT signals are known to be involved in the maturation, cytotoxicity, or survival of NK cells [23,24,25,26]. 

This study aims to investigate the immunosuppressive mechanisms that could affect NK cell function in the thyroid cancer microenvironment. Previous studies have examined with supernatant of papillary thyroid cancer (PTC) cell line and anaplastic thyroid cancer (ATC) cell lines to examine the differences in prognosis of thyroid cancer [27]. We examined the function of NK cells in co-culture with the thyroid cancer cells, to elucidate the factors and mechanisms responsible for the suppression of NK cell function and NK receptor expression. To date, restricted information is available on the effect of thyroid cancer cells on NK cells. These results could provide helpful tips for the development of effective cancer therapeutics that can improve NK cell function. 

## 2. Experimental Section

### 2.1. Cell Lines and Culture

Thyroid cancer cell lines, including TPC-1, FRO, and 850-5C were purchased from American Type Culture Collection (ATCC, Manassas, VA, USA). FRO and 850-5C cells were cultured in Roswell Park Memorial Institute (RPMI) containing 10% FBS (Welgene, Gyeongsan, Korea). TPC-1 cells were cultured in Dulbecco’s Modified Eagle Medium (DMEM) containing 10% Fetal Bovine Serum (FBS). In co-culture experiments, NK cells were cultured for 24 h with thyroid cancer cell lines in IL-2 (10 ng/mL) at a thyroid/NK ratio of 1:5 (thyroid cancer cells cultured 2 × 10^5^ cells and NK cells cultured 1 × 10^6^ cells per 6 well). 1-methyl-DL-tryptophan (1MT, an IDO inhibitor; Sigma-Aldrich, St. Louis, MO, USA) was added at 2 mmol/L (2 mM) during co-incubation for 24 h. A total of 2 μM of NS398 (PGE2 inhibitors, Sigma-Aldrich) or anti- TGF-β (Sigma-Aldrich) was added to the co-culture system. When indicated, the following specific reagents were added to NK cells; L-kynurenine, CH223191, JSI-124, Fludarabine (Sigma-Aldrich). Also, 293T cells were purchased from American Type Culture Collection (ATCC, Manassas, VA, USA) and were cultured in DMEM containing 10% FBS. NK cell lines including NK92 and NK leukemia (NKL) were obtained from American Type Culture Collection (ATCC, Manassas, VA, USA). NK92 cells were cultured in α-MEM containing 0.2 mM inositol, 0.1 mM 2- mercaptoethanol, 0.02 mM folic acid, 100 U/mL recombinant IL-2, 12.5% horse serum, 12.5% FBS. NKL cells were cultured in RPMI in the presence of 15% FBS, 2 mM L-glutamate, 100 μg/mL penicillin, 100 μg/mL streptomycin, and 100 IU/mL IL-2. 

### 2.2. NK Cell Isolation and Culture

Human NK cells were isolated from peripheral blood (PB) of a healthy donor. Rosette Sep (StemCell Technologies) was added into PB for elimination of CD3+ cells and red blood cells. Then NK cells were isolated using CD56 magnetic beads (Miltenyi Biotec, Bergisch Gladbach, Germany). A total of 1 × 10^6^ CD56 cells per 24 well were cultured in α-minimal essential medium (Welgene) supplemented with human IL-15 (10 ng/mL), IL-21 (10 ng/mL), and 10^−6^ M hydrocortisone (StemCell Technologies). All cytokines used for NK cell culture were purchased from PeproTech (Rocky Hill, NJ, USA). This study was approved by the Institutional Review Board of the Asan Medical Center according to the Bioethics and Safety Act and the Declaration of Helsinki. Each participant provided written, informed consent (Ethical code number: 10141091).

### 2.3. Evaluation of NK Cell Cytotoxicity

Cytotoxicity was evaluated using a calcein-AM release assay [28]. Briefly, target cells were labeled with calcein-AM (Invitrogen, Carlsbad, CA, USA) for 1 h. Calcein-labeled target cells (1 × 10^4^ cells per well) and serially diluted effector cells were then co-cultured in 96-well round-bottom plates for 4 h. “Maximum release” was simulated by adding 2% Triton X-100 to the target cells, and “spontaneous release” was simulated by adding culture media to the target cells. The calcein released into the supernatant was measured using a multi-mode microplate reader (Molecular Devices, San Jose, CA, USA). The percent specific lysis was calculated according to the formula ((test release-spontaneous release)/(maximum release-spontaneous release)) × 100.

### 2.4. Flow Cytometry Analysis

NK cells were stained with indicated antibodies for further analysis. For immunostaining, the cells were washed two times with phosphate-buffered saline containing 2% fetal bovine serum, adjusted to approximately 1 × 10^5^ to 1 × 10^6^ cells in 100 μL of the same buffer, and labeled with FITC-anti-CD158a, PE-anti-CD158b, BV-anti-CD56, APC-anti-NKp46, PE-anti-NKp44, FITC-anti-CD16, APC-anti-NKp30, FITC-anti-CD158e, PE-anti-NKG2D, PE-anti-TRAIL, or PE-anti-AhR. Incubations with antibodies were performed for 25 min at 4 °C in the dark. Antibodies for immunostaining were purchased from BD Bioscience. The data of samples were acquired by Canto II (BD Bioscience, Franklin Lakes, NJ, USA) and analyzed using software Flow Jo (Tree Star, Inc., Ashland, OR, USA).

### 2.5. IFN-γ and Kynurenine Measurement 

Interferon-γ (IFN-γ) was evaluated in the NK cell supernatant obtained from the co-culture with thyroid cancer cells by specific enzyme-linked immunosorbent assay (ELISA) kit purchased from eBioscience (Waltham, MA, USA). Kynurenine was measured in the thyroid cancer cell supernatant obtained from the co-culture with NK cells. A total of 60 μL of sample/standard and 30 μL of 30% trichloroacetic acid (Sigma-Aldrich, St. Louis, MO, USA) were incubated for 30 min at 50 °C in 96-well round-bottom culture plate to hydrolyze N-formyl kynurenine to kynurenine. The 96-well plates were centrifuged at 3000× *g* for 10 min and 70 μL of supernatant was obtained. Equal amounts of Ehrlich Reagent (2% p-dimethylaminobenzaldehyde in glacial acetic acid) were added to the supernatants for reaction. Absorbance was read at 492 nm. 

### 2.6. Western Blot Analysis 

To measure IDO levels in thyroid cancer cells, aliquots of 5 × 10^5^ cancer cells were incubated at 37 °C for 48 h untreated or treated with IFN-γ 10 ng/mL or co-cultured with NK cells (1 × 10^6^). The thyroid cancer cells were treated with 1 or 2 mM of 1 MT for blocking the IDO expression stimulated by IFN-γ. Cell lysis was carried out by radioimmunoprecipitation using assay cell lysis buffer (GenDEPOT, Katy, TX, USA) with protease inhibitor. Samples were separated by 9% Sodium Dodecyl Sulfate Polyacrylamide Gel Electrophoresis (SDS–PAGE) and transferred onto 0.45 μm-pore polyvinylidene difluoride membranes (Millipore, Bedford, MA). After 1 h of blocking in PBS supplemented with 0.05% Tween 20 (Duchefa Biochemie, NH, Netherlands) containing 5% skimmed milk at room temperature, the membranes were incubated overnight with primary antibodies at 4 °C. The primary antibodies used were β-actin (Santa Cruz Biotechnology, CA, USA) or IDO (Cell Signaling Technology, Danvers, MA, USA). Subsequently, the membranes were incubated with corresponding Horseradish peroxidase (HRP) conjugated anti-rabbit, anti-mouse antibody (Santa Cruz Biotechnology, CA, USA) for 1 h at room temperature.

For NK signaling pathway analysis, 1 × 10^6^ NK cells were cultured with indicated concentrations of kynurenine at 37 °C for 24 h and then lysed in lysis buffer. 293T and NK cell lines including NK 92 and NKL were cultured in a condition media (2 × 10^5^ to 5 × 10^5^ cells per 6-well plates). Primary antibodies against STAT1 (42H3), phosphorylated (p-) STAT1, STAT3 (124H6) and p-STAT3 were purchased from Cell Signaling Technology.

The Western blot bands were detected with luminol/enhancer solution and stable peroxide solution (Thermo Fisher Scientific, MA, USA). The intensity of each band was obtained using the program CSAnalyzer 4 (ATTO Technology, NY, USA) and normalized to β-actin. Fold change was used to compare the relative abundance of a target protein to the control sample on the same membrane.

### 2.7. Quantitative Real-Time PCR

Total RNA was extracted using the RNeasy^®^ Mini kit (Qiagen, Hilden, Germany) according to the manufacturer’s instructions. Total RNA was reverse-transcribed using cDNA synthesis kit (Toyobo, Osaka, Japan), and real-time PCR was performed in a Dice TP 800 Thermal Cyclear with SYBR^®^ Premix (Takara Co., Shiga, Japan). Real-time PCR reactions were carried out in a 18 μL volume containing 10 pmol/μL primers and 1 μL cDNA using the following conditions: one cycle of 95 °C for 30 s, 40 cycles of 95 °C for 5 s, and 60 °C for 10 s; and a dissociation stage of 1 cycle at 95 °C for 15 s, 60 °C for 30 s, and 95 °C for 15 s. Results were normalized to the housekeeping genes *GAPDH*. Gene expression values were calculated with the 2^−ΔΔCt^ method. Relative quantification of gene expression was determined by comparison of fold value. The primer sequences were as follows; for AhR 5′-CAACAGCAACAGTCCTTGGC-3′ and 5′-GTTGCTGTGGCTCCACTACT-3′; for GAPDH 5′-GCACCGTCAAGGCTGAGAAC-3′ and 5′-TGGTGAAGACGCCAGTGGA-3′. 

### 2.8. Luciferase Assay

To generate NKG2D promoter-pGL3 or NKp46 promoter-pGL3, human NKG2D and NKp46 promoter sequences were amplified from human NK cell genomic DNA by PCR with specific primers. Promoter primers were designed using Takara in-fusion-cloning-tools (https://www.takarabio.com). The sequences were as follows: For NKp46 5′-TTTCTCTATCGATAGGTTGGGACTACAGGCATGTGC-3′ and 5′-CCGGAATGCCAAGCTCGCTCAGATTCTGCCGGC-3′; for NKG2D 5′-TTTCTCTATCGATAG GGTCAATGGGTACAAAGT-3′ and 5′-CCGGAATGCCAAGCTAATAATGTAAAGATTTAAAAATAGT-3′. The products were cloned into the BamHI and AvaI restriction enzyme sites of the pGL3-basic vector (Promega, Madison, WI, USA). STAT1 and STAT3-pOTB7 recombined vectors were provided from Korea Human Gene Bank (Medical Genomics Research center, KRIBB, Korea). A total of 2 × 10^5^ 293T cells were seeded in the 24-well plates and were transfected the next day with Lipofectamine and PLUS reagent (Invitrogen™, Waltham, MA, USA) according to the instructions provided by the manufacturer. Recombinant pGL3 plasmids or pGL3-basic and STAT-OTB7 plasmids were transfected into the corresponding wells and pRL-TK vector containing the *Renilla* luciferase gene as an internal control was added to each well. The cells were lysed in standard 1× lysis buffer and the cell lysates were assayed for both firefly and *Renilla* luciferase activity using the luciferase reporter assay kit (Promega) according to the instructions provided by the manufacturer.

### 2.9. Statistical Analysis

Statistical significance was evaluated by Student’s *t*-test using GraphPad prism software and Microsoft Excel. A *P* value of less than 0.05 (*), less than 0.01 (**), or less than 0.001 (***) was considered statistically significant.

## 3. Results

### 3.1. Thyroid Cancer Cells Inhibit NK Cell Cytolytic Function and NK Receptor Expression

NK cells were collected and analyzed after co-culture with thyroid cancer cells. The cytolytic function of NK cells decreased after co-culture with thyroid cancer cells, even though the level was depended on the thyroid cancer cells in the co-culture (Figure 1A,B). The percentage of positive cells expressing NK cell receptors especially activating receptors such as, NKp46, CD16, NKp30, and NKG2D, also decreased after co-culture. The expression of the death receptor TRAIL was also significantly decreased (Figure 1C). These results indicate that the interaction with thyroid cancer cells resulted in reduced NK activating receptors, which suppressed NK cell cytotoxicity. 

### 3.2. IDO Expression in Thyroid Cancer Cells is Induced by Co-Culture With NK Cells 

Previous reports showed that soluble factors from cancer cells could inhibit NK cell function [29,30,31]. We examined changes in NK cell function by inhibiting each factor (TGF-β or PGE2 or IDO) in the co-culture environment using their inhibitors (Appendix A). Only the inhibitor of IDO partially restored NK cell function. IDO is one of the factors known to be expressed by many cancers to aid their escape from the immune system. First, we examined NK cell cytotoxicity during treatment with an IDO inhibitor (1MT) in a co-cultured environment. The cytolytic activity of NK cells was decreased when NK cells were cultured with thyroid cancer cells, but it partially restored when NK cells were co-cultured with thyroid cancer cells and treated with 1MT (an IDO inhibitor). ATC-derived cells FRO and 850-5C similarly decreased NK activity and were affected by IDO inhibitors. (Figure 2A and Appendix A). Treatment with 1MT also restored NK receptor expression in NK cells, which was reduced by co-culture with thyroid cancer cells (Figure 2B). IDO was not expressed in thyroid cancer cells cultured alone, but its expression was induced by treatment with IFN-γ or co-culture with NK cells (Figure 2C). IFN-γ was produced in the co-culture of NK cells and thyroid cancer cells (Figure 2D), which indicates that IFN-γ produced in co-culture conditions induces IDO expression in thyroid cancer cells. As shown Figure 2E, IDO expression was not induced during treatment with 1MT, despite the stimulation of IFN-γ. In addition, we examined the production of kynurenine, a metabolite of IDO, known to downregulate NK cell function in culture supernatant. Its levels were very low when thyroid cancer cells were cultured alone, but significantly increased when they were cultured with NK cells (Figure 2F). These results suggest that thyroid cancer cells reduced NK cell cytolytic activity and expression of NK activating receptors by producing kynurenine using IDO.

### 3.3. Kynurenine Inhibits NK Cell Activity via STAT Signaling Pathways.

We treated NK cells with kynurenine and observed the mechanism responsible for its reduction of NK cell function. NK cells were treated with different concentrations of kynurenine, and their cytolytic activity against K562 was measured. The cytolytic activity of NK cells was decreased in a dose-dependent manner (Figure 3A). NK cells treated with kynurenine showed decreased cytolytic activity against thyroid cancer cells as well (Appendix A). The expression of activating receptors by NK cells was also decreased, a similar pattern to that obtained when NK cells were co-cultured with thyroid cancer cells. The expression of NKp46 and NKG2D tended to decrease as the concentration of kynurenine increased, compared with the expression of the other receptors (Figure 3B). Next, we performed Western blotting for analysis of transcription factors known to be involved in NK cell activation, to identify the signaling pathway responsible for NK cell function decrease by kynurenine. Treatment with kynurenine dose-dependently decreased the quantity of the phosphorylated forms of STAT3 and STAT1 (Figure 3C,D), but not the activation of the other factors associated with NK activity such as NF-κB (nuclear factor kappa-light-chain-enhancer of activated B cells), p38 and ERK (extracellular-signal-regulated kinase) (Appendix A). These results indicate that kynurenine produced by IDO in thyroid cancer cells inhibits the function of NK cells by downregulating STAT3 and STAT1 activation, which may also be related to the expression of NKp46 and NKG2D.

### 3.4. Kynurenine Affects NK Cell Activity Through the Aryl Hydrocarbon Receptor (AhR)

We examined the pathway through which kynurenine enters NK cells. First, we checked mRNA expression and surface expression of AhR, known as the kynurenine receptor in NK cells (Figure 4A,B). AhR expression was detected in NK cells and AhR mRNA levels increased in kynurenine treatment, though not in a dose-dependent manner. However, kynurenine treatment did not affect cell protein expression. We used the AhR antagonist CH223191 to determine if kynurenine affects NK cells by using AhR expressed on NK cells. The cytolytic activity of NK cells, and expression of NKG2D and NKp46 were restored by pretreatment with 1 μM or 10 μM CH223191 (Figure 4C,D). The changes in NK cells resulting from kynurenine inhibition by CH223191 were analyzed. Western blot analysis revealed that the signals for the activation of STAT3 and STAT1 were blocked when NK cells were treated with kynurenine. Also, blocking of AhR resulted in the activation of the STAT3/1 pathway (Figure 4E and Appendix A). These results indicate that kynurenine decreases the cytolytic function of NK cells and activation of STAT signals via the AhR expressed on NK cells. 

### 3.5. STATs Regulate NK Cytolytic Activity and Receptor Expression.

Inhibitors of STAT3 (JSI-124) and STAT1 (fludarabine) signaling were used to investigate their roles in NK cell function decrease. The results showed that they decreased the cytolytic activity of NK cells. Co-treatment resulted in a larger decrease than monotherapy (Figure 5A), similar to the observation made after treatment with kynurenine (Figure 5B). Treatment with STAT inhibitors also resulted in decreased NKp46 and NKG2D expression (Figure 5C), as similarly observed in kynurenine treatment. These results confirmed that kynurenine regulates NK cells by using STAT1 and STAT3 signaling to modulate the expression of NK receptors such as, NKp46 and NKG2D.

### 3.6. Direct Involvement of STATs in NK Receptor Expression

NK receptor expression has been implicated in NK cell function. We assumed that the decreased expression of NK cell receptors such as, NKG2D and NKp46, resulted in the decreased NK cell function induced by kynurenine. Thus, we examined the effect of STAT1 and STAT3 regulated by kynurenine on NK cell receptor expression. We found that STAT1 and STAT3 mediate gene modulation effects by binding to specific target sequences in NK receptors. First, we checked the presence of the STAT1 and STAT3 binding motifs (TTCN_2–4_GAA) in the promoter sequence of NK receptors using the program Find Individual Motif Occurences (FIMO) (http://meme-suite.org/ tools/fimo), to confirm the possibility that the STATs regulate the expression of NK receptors. We obtained a sequence of promoters from the University of California Santa Cruz (UCSC) page (https://genome.ucsc.edu/), and were able to verify the presence of STATs motifs (Appendix A). A pGL3 vector in which the promoter of each NK cell receptor containing the STAT binding motif was cloned, was used for the experiments. 293T cells were also used for the experiment because they were not only highly efficient in transfection, but also had low activation form of STAT3 and STAT1 (Figure 6A). To confirm the direct influence of STATs on NK cell receptor expression, a pGL3-NKp46 or pGL3-NKG2D vector containing the luciferase gene under the control of about 1-kb NKp46 or NKG2D promoter was transfected into 293T cells and luciferase activity was measured after 48 h. At this time, a vector and a STAT1 or STAT3 clone were transfected with a promoter vector to stimulate a STAT signal, and a basic pGL3 vector was used as a control. Both STAT1 and STAT3 signals enhanced transcription of NKp46 and NKG2D. STAT1 showed a higher response to NKp46, and STAT3 showed a higher response in NKG2D (Figure 6B). These results indicate that the phosphorylated forms of STATs directly bind to the promoters of NK receptors (NKG2D or NKp46), which regulates the expression level of the receptor.

## 4. Discussion

In this study, we demonstrated that thyroid cancer cells affect the function of NK cells by regulating indoleamine 2,3-dioxygenase (IDO) expression through the production of kynurenine. The expression of NKG2D and NKp46 receptors was especially downregulated by kynurenine treatment via STAT signaling pathways. In addition, we have newly found that STAT1 and STAT3 directly bind to the NKp46 and NKG2D promoters and regulate their expressions. The cytolytic ability of NK cells was decreased by co-culture with thyroid cancer cells. In particular, NK cells used in this study showed highly expressed activating receptors by stimulating cytokines such as IL-15, IL-21, and IL-2, but the expression of NK cell receptors decreased after co-culture. This may have resulted from the influence of several factors from the cancer microenvironments [9,29,32]. In our previous study, we investigated factors in thyroid cancer cell culture supernatant that could indirectly affect NK cells and found that PGE2 affected NK cells [27]. However, in the present study, treatment with PGE2 inhibitors had no effect on the reduced NK cell function induced by co-culture with thyroid cancer cells, suggesting that there may be other factors involved than PGE2 (Appendix A); however, due to its ability to break down the essential amino acid tryptophan into kynurenine, its ability to aid immune system regulation of tumor growth has extensively been studied [33,34,35]. In addition, it has recently been reported that IDO catabolites block the proliferation of T and NK cells [36,37]. In the present study, blocking IDO in the co-culture system using its inhibitor 1MT, restored NK cell cytolytic activity and receptor (particularly NKp46 and NKG2D) expression. IDO production by thyroid cancer cells was induced by stimulation with IFN-γ or co-culture with NK cells, and kynurenine production was also increased by co-culture with NK cells. Since kynurenine is known to directly affect many immune cells [38,39,40], we investigated if it directly affects NK cells. When NK cells were treated with kynurenine, NK cell receptor expression and cytolytic activity decreased (Figure 3A,B). The expression of NKp46 and NKG2D receptors was decreased by kynurenine, consistent with NK receptor expression decrease observed after co-culture with thyroid cancer cells, and they were both restored by treatment with the IDO inhibitor (Figure 2). These suggest that kynurenine produced by IDO directly impairs NK cell function. Kynurenine treatment led to the decreased phosphorylation of STAT1 and STAT3 in NK cells (Figure 3C). It has been reported that STAT signals are activated by IL-15, 21, and IL-2 in NK cells [41]. We found that kynurenine treatment reduced the expression of P-STATs in NK cells in a dose-dependent manner, indicating that kynurenine regulates NK cells via STATs signaling pathways. In addition to STAT1 and STAT3, we examined the change of other STAT family expression level by kynurenine. We examined the expression of STAT4 that is known to be binding to the promoter of IFN-γ, and the expression of STAT5 that is involved in the development and function of immune suppression of NK cells [23,42]. However, the expressions of STAT4 and 5 were not significantly changed by the treatment of kynurenine (Appendix A).

Kynurenine has been known to enter cells through the aryl hydrocarbon receptor (AhR). AhR is activated or inactivated by ligands including kynurenine, and many chemical ligands. AhR acts as a transcription factor to control metabolic enzymes and is also known to be involved in immunity, stem cell maintenance, and cell differentiation [43,44,45]. Treatment with the AhR antagonist CH223191 recovered the impaired NK cell function induced by kynurenine treatment (Figure 4C). CH 223191 is an effective 2, 3, 7, 8-Tetrachlorodibenzo-p-dioxin (TCDD)-induced AhR inhibitor, that blocks the binding of TCDD to AhR, and prevents TCDD mediated nuclear transfer and AhR DNA binding [46,47]. These results demonstrate that kynurenine enters the NK cell via AhR. In addition, downregulation of STATs activation was restored by treatment with an AhR antagonist, which confirms that kynurenine regulates the STAT pathways through AhR (Figure 4).

In some cancer cells, STAT3 is known to escape NK cell immune surveillance by regulating NKG2D ligand expression. In addition, STAT3 plays a pivotal role in carcinogenesis and immunosuppression [48,49]. In contrast, recently, STAT3 signaling has been reported to promote the proliferation and function of human NK cells activated by IL-21 stimulation. In this study, we have elucidated a mechanism that regulates STAT pathways using IDO in NK cells, other than previously reported mechanisms that use cytokines. STAT signals were activated in NK cells used in this study, which could be caused by the cytokines lL-21 and IL-15 detected in the NK cell culture medium (Figure 3C and Figure 4E). We also showed the relationship between STAT and NKp46, and STAT and NKG2D. Both STAT1 and STAT3 signals enhanced transcription of NKp46 and NKG2D, with STAT1 showing a higher response to NKp46, and STAT3 showing a higher response in NKG2D. We also found that the cytolytic activity of NK cells was decreased by treatment with STAT1 and STAT3 inhibitors (Figure 5 and Figure 6). By performing promoter assay, we confirmed that STATs bind to the promoter of NKG2D and NKp46 to activate transcription (Figure 6). The same palindromic core motif TTCN_2–4_GAA, was found in sequences recognized by all STATs [50,51,52]. Thus, activation of STAT3/1 in NK cells regulate receptor expression by direct binding to promoter of NK activating receptors.

## 5. Conclusions

In conclusion, IDO expression in thyroid cancer cells induced by interaction with NK cells in a tumor microenvironment may protect themselves from immune system surveillance through the suppression of NK cell function by producing kynurenine. Kynurenine decreases the cytolytic activity of NK cells, and NKG2D and NKp40 expression by downregulating the activation of the STAT1 and STAT3 pathways in NK cells. These data present the possibility that modulation of IDO could improve NK cell function in thyroid cancer therapy.

## Figures and Tables

**Figure 1 jcm-08-00842-f001:**
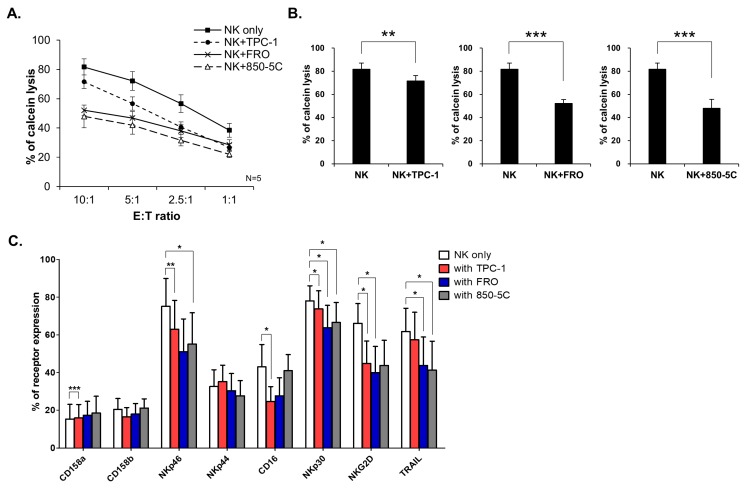
Thyroid cancer cells inhibit natural killer (NK) cell cytotoxicity and NK receptor expression. (**A**) NK cells were cultured without thyroid cancer cells (square) or co-cultured with TPC-1 (circle), FRO (cross), and 850-5C (triangle) in the medium containing IL-2 for 24 h. The cytotoxicity of NK cells was assessed against K562 cells at different E:T ratio as described in Materials and Methods. Bars represent mean ± SD from five experiments. (**B**) Graphs of NK cell cytotoxicity cultured with each thyroid cancer cells relative to NK cells grown in control medium were shown. K562 cells were used as targets and the effector: target (E: T) ratio was 10:1. Bars represent mean ± SD of five independent experiments. Statistical analyses were performed using the paired two-tailed Student’s *t*-test. (**C**) Expressions of NK receptors were analyzed by flow cytometry. Graph indicates that the percent of NK receptor expressions in CD56 positive cells. Bars represent mean ± SD of six independent experiments. Statistical analyses were performed using the paired two-tailed Student’s *t*-test. * *P* < 0.05, ** *P* < 0.01, and *** *P* < 0.001.

**Figure 2 jcm-08-00842-f002:**
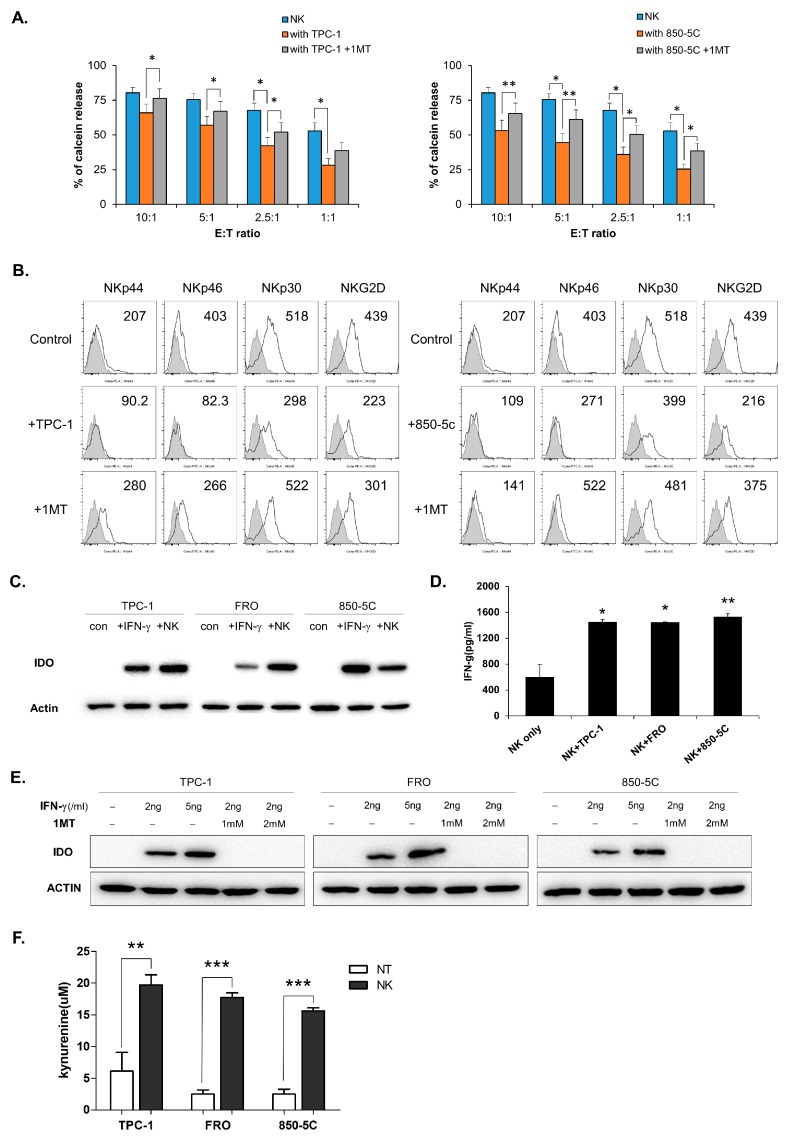
Indoleamine 2,3-dioxygenase (IDO) inhibitor, 1MT (1-Methyl-D-tryptophan) blocks IDO expression and restores NK cell activity reduced by thyroid cancer. (**A**) NK cells were cultured for 24h with IL-2 either alone (blue) or with the indicated thyroid cancer cells. NK/thyroid cell co-cultures were set either in the absence (orange) or in the presence (gray) of IDO inhibitor. Bars represent mean ± SD of five independent experiments. (**B**) Expressions of NK receptors, NKp44, NKp46, NKp30, and NKG2D (white profiles) and isotype controls (gray filled profiles) were analyzed by flow cytometry. Numbers indicate mean fluorescence intensity (MFI) and Y-axis shows cell counts. Control indicated NK cultured alone and +TPC or +850-5C indicated NK cells cultured with thyroid cancer cells. +1MT indicated NK cells cultured with thyroid cancer cells in the presence of IDO inhibitor. (**C**) IDO expression was detected by Western blotting in thyroid cancer cells; untreated thyroid cancer cells, IFN-γ-treated thyroid cancer cells and thyroid cancer cells co-cultured with NK cells. β-actin served as a loading control. (**D**) IFN-γ concentration in culture supernatants from NK cell co-cultured with the indicated thyroid cancer cells. Bars represent mean ± SD of three independent experiments. Statistical analyses were performed in comparison with NK only group using the paired two-tailed Student’s *t*-test. (**E**) The expression of IDO enzyme was assessed by Western blot analysis on untreated, IFN-γ-treated thyroid cancer cells with 2 ng or 5 ng per m, 1MT-treated thyroid cancer cells with 1 mM or 2 mM. (**F**) L-kynurenine concentration was measured in culture supernatants from thyroid cancer cells only (NT, white), and thyroid cancer cells co-cultured with NK cells (dark gray). Bars represent means ± SD obtained from six independent experiments. Statistical analyses were performed using the paired two-tailed Student’s *t*-test. * *P* < 0.05, ** *P* < 0.01, and *** *P* < 0.001.

**Figure 3 jcm-08-00842-f003:**
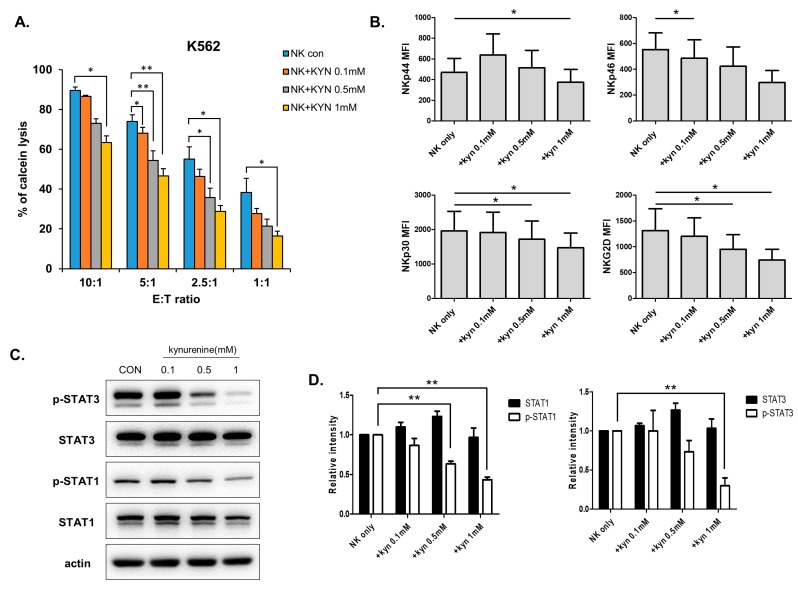
Kynurenine directly inhibits NK cell activity and NK receptor expressions. (**A**) NK cells were cultured with IL-2 in the absence (blue) or presence of 0.1 mM (orange) or 0.5 mM (gray) or 1mM (yellow) of L-kynurenine for 24 h and cytotoxicity against K562 cells was examined. Bars represent mean ± SD of three independent experiments. (**B**) Expressions of NK receptors, including NKp44, NKp46, NKp30, and NKG2D were detected by flow cytometry. Graphs represent mean values of mean fluorescence intensity (MFI) for three independent experiments. (**C**) Phosphorylation of signal transducer and activator of transcription (STAT)1 and STAT3 were analyzed by Western blotting after the treatment of kynurenine in NK cells. β-actin served as a loading control. (**D**) Relative intensity is defined as the intensity of the target protein normalized to β-actin. Bars represent mean ± SD of three independent experiments. Statistical analyses were performed using the paired two-tailed Student’s *t*-test. * *P* < 0.05 and ** *P* < 0.01.

**Figure 4 jcm-08-00842-f004:**
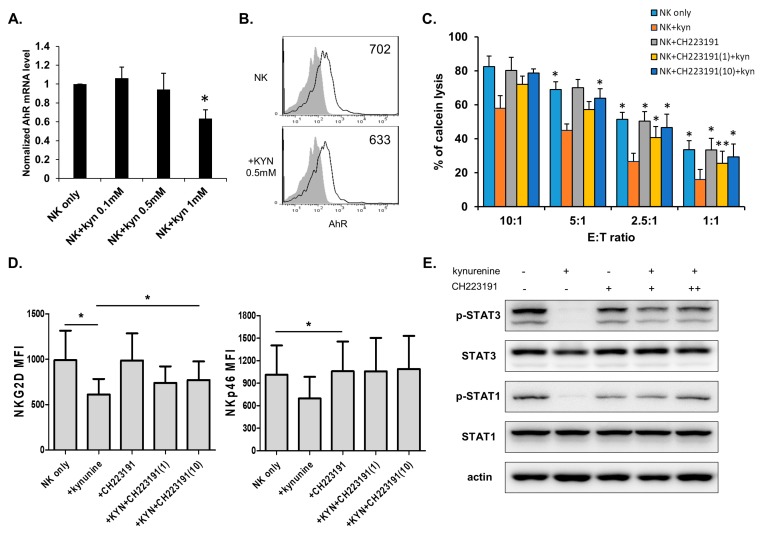
Kynurenine inhibits NK cell activity through aryl hydrocarbon receptor (AhR). mRNA level (**A**) and surface expression (**B**) of AhR were analyzed in kynurenine-treated or untreated NK cells. (**C**) NK cells were cultured with IL-2 in the absence (blue) or presence of indicated component for 24 h and cytotoxicity against K562 cells was examined; L-kynurenine (orange), CH223191(gray), kynurenine with CH223191 (yellow and dark blue). Bars represent mean ± SD of three independent experiments. (**D**) Expressions of NK receptors, NKG2D, and NKp46 were analyzed by flow cytometry. Graphs represent mean values of MFI for three independent experiments of receptors. (**E**) Phosphorylation of STAT-3 and 1 signals were examined by Western blotting in NK cells under the indicated conditions. β-actin served as a loading control. Statistical analyses were performed using the paired two-tailed Student’s *t*-test. * *P* < 0.05 and ** *P* < 0.01.

**Figure 5 jcm-08-00842-f005:**
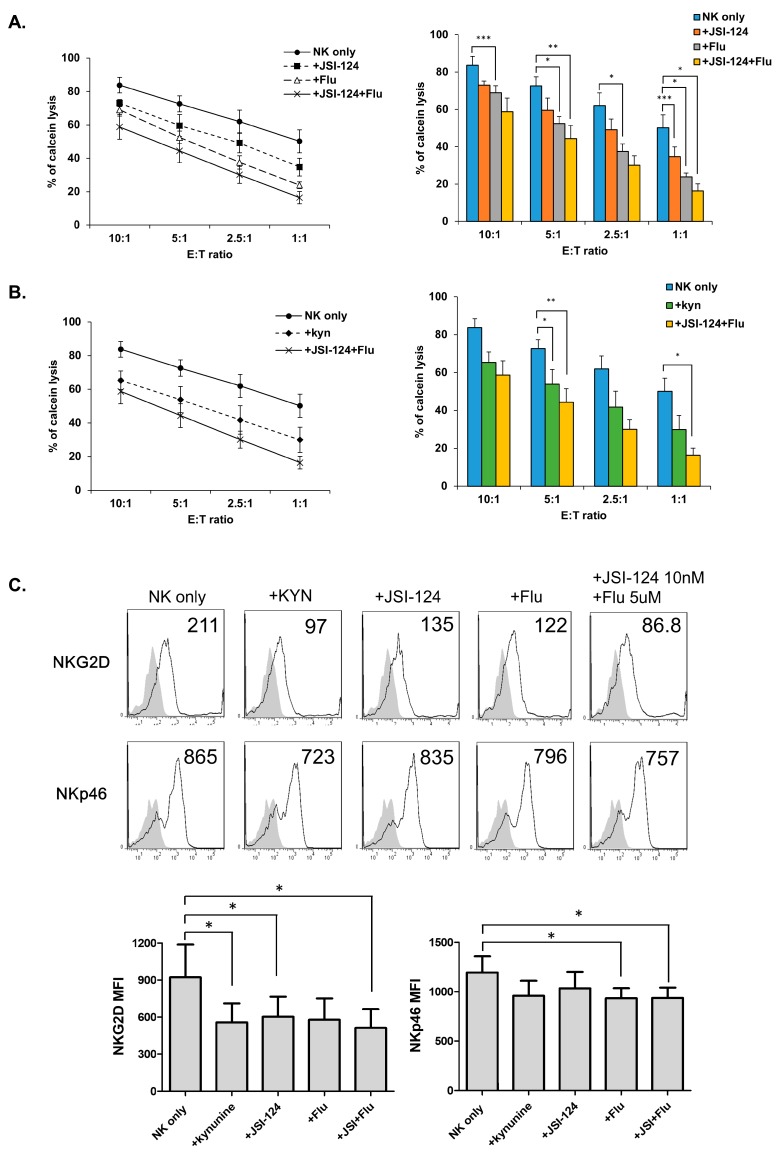
STATs signals regulate the activity of NK cells. (**A**) NK cells were cultured for 24 h with IL-2 either alone (left- circle, right- blue) or in the presence of the indicated inhibitor of STATs; STAT3 inhibitor (JSI-124, left-square, right-orange) or STAT1 inhibitor (fludarabine, left-triangle, right-gray) or both of them (left-cross, right-yellow). After 24 h, the cytotoxicity of NK cells against K562 cells was examined (**B**) NK cells were treated with STAT inhibitors or kynurenine for 24 h, and the cytotoxicity of NK cells against K562 was evaluated; circle indicates untreated NK cells (blue), diamond indicates NK cells treated with kynurenine (green) and cross indicates NK cells treated with STAT inhibitors (fludarabine+JSI-124, yellow). Bars represent mean ± SD of three independent experiments. (**C**) Expression of NKG2D, NKp46 (white profiles), and isotype controls (gray filled profiles) were analyzed by flow cytometry. Numbers indicate mean fluorescence intensity and Y-axis shows cell counts. The graphs below represent mean values of MFI for four independent experiments. Statistical analyses were performed using the paired two-tailed Student’s *t*-test. * *P* < 0.05, ** *P* < 0.01, and *** *P* < 0.001.

**Figure 6 jcm-08-00842-f006:**
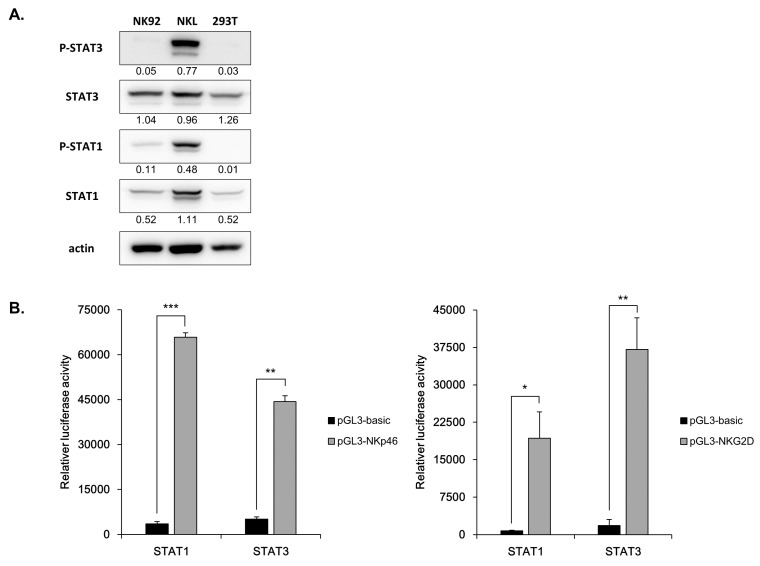
Expressions of NKG2D and NKp40 are directly regulated by STATs. (**A**) STAT 1 and STAT3 expressions were measured by Western blotting in indicated cell lines. The numbers below each band indicated quantitative analysis of target protein relative to β-actin. (**B**) 293T cells were transfected with the indicated vectors. After 48 h, luciferase activity was quantified according to the manufacturer’s protocol. The efficiency of transfection was calibrated to the Renila value. Bars represent mean ± SD of three independent experiments. Statistical analyses were performed using the paired two-tailed Student’s *t*-test. * *P* < 0.05, ** *P* < 0.01, and *** *P* < 0.001.

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
