# Peer review of "Indoleamine-2,3-Dioxygenase in Thyroid Cancer Cells Suppresses Natural Killer Cell Function by Inhibiting NKG2D and NKp46 Expression via STAT Signaling Pathways"

_jcm, 2019, doi:10.3390/jcm8060842_

Round 1

Reviewer 1 Report

In this interesting manuscript, the authors investigated a possible mechanism that could affect NK cell function in the thyroid cancer microenvironment. In a co-culture system, thyroid cancer cells downregulate NK cells cytotoxicity and the expression of activating receptors, such as NKG2D and NKp46. This inhibiting effect is due to kynurenine produced by thyroid cancer cells metabolizing IDO. Kynurenine enters in NK cells via AhR, where it decreases NK cell function and receptors expression activating STAT1 and STAT3 pathways. They found that STAT1 and STAT3 directly bind promotor regions of NKG2D and NKp46 receptor modulating their transcription.

Overall, the experiments are done in a methodically and the results are very interesting. However, I have some minor concerns about the presentation of the data. Authors presented results in a mainly descriptive way. I think they should always show quantifications and statistical significance, that would only improve the significance of their findings.

These are some specific points need to be addressed by the authors:

1. I think some references are missing throughout the text.

3. Material and Methods section should be written with more details:

- Authors should specify how many cells they used in each experiment.

- Western blot analysis should be written extensively. What they used to run gels? how they transfer the gels, in which kind of membrane? Which secondary antibody they used? How they developed membranes?

- Also, in the description of quantitative Real-Time PCR some details are missing, for example it is not clear which method they used to analyze data: fold change? relative expression procedure?

- In the description of luciferase assay they should provide primers used to amplify promoter regions of human NK2GD and NKp46. How many HEK293 have been seeded and used?

- Which analysis tool have been used to performed statistical analysis?

3. Results. As I said above, results are mainly descriptive. Authors should perform quantification of all the data they present, and they should show all p values throughout the text. Please perform statistical analysis for all the data (Figure 2A, 2B, 3A, 3B, 3C, 4A, 4B, 4C, 4D, 4E, 5A, 5B, 5C, 6A).

- Page 4, line 147 “ thyroid cancer cell unit inhibit NK cell cytolytic function and NK receptor expression” These results are really similar to the results published by the same authors in Front Immunol. 2018; 9: 1859, and I don’t think these results add much more.

- Page 4, lines 150: authors said that in co-culture system with cancer cells, NK cell function was decreased. Here authors should explain which thyroid cancer cells have been used (not only in material and methods section) and why they used these specific thyroid cell lines.

Moreover, NK cell function decreased but the level depends on the thyroid cancer cell. Could the authors provide a hypothesis for these differences?

-Page 4 and page 6, there is a reason why authors showed receptor expression data plotted in two different ways? E.g Figure 1C and Figure 2B. I think data plotted as in Figure 1C are clearer, moreover the quantification with p values should be provide.

- Page 5, line 169. Authors used inhibitors factor for TGF-b, PGE and IDO, but they don’t explain it in material and method section.

Furthermore, authors treated TPC-1 and 850-5c cell lines with IDO-inhibitor, but not FRO cell line. They should show data also for FRO cells, both for NK cytolytic activity and NK receptor expression.

- Page 5, lines 171 and 174. Authors said that treatment with the inhibitor of IDO restored NK cell function. However, data do not support this affirmation. It is more correct to say “partially restored” (as they said in line 174), but still they could not assume it until they provide a quantification with relative statistical analysis.

-Page 5, authors should explain why they treated cells with IFN-gamma.

- Page 6, Fig. 2B. What “Con” stands for? I think “control” however  authors should explain it at least in the figure legend.

- Page 7-8, again, provide quantification for western blot analysis (Figure 1C).

-Page 8, in Figure 4A authors showed AhR mRNA expression in untreated and kynurenine-treated cells. Please provide error bars and statistical significance.

-Page 8 in Figure 1B lower panel, please specify the concentration of kynurenine used for NK treatment and statistical significance.

- Page 8, Figure 4E. Please provide quantification and statistical significance of western blot analysis.

-Page 9, Figure 5. Again, for all panels statistical analysis should be provide.

-Page 10, Figure 6A. In this western blot authors analyzed phosphorylated forms of STAT1 and STAT3 in 3 different cell lines. Please, present quantification of western blot analysis and described these cell lines in the material and methods section.

- Page 10, lines 10. Which software authors used to verify the presence of STAT1 and STAT3 binding motifs in the promoter region of NK receptors NKp46 and NKG2D?

- Page 10, The results of this luciferase assay are interesting. Nevertheless, authors should perform this experiment on a thyroid cell line.

Author Response

Response to reviewer 1 comments

1. I think some references are missing throughout the text.

Response 1: We put some references in the text as follows;

- reference # [5] in page 1, line 37

- reference # [13] in page 2, line 51

We also added references with new text

- reference # [10] page 2, line 46

- reference # [42] page 14, line 384

2. Material and Methods section should be written with more details:

: As you suggested, we added more detailed description in Materials and Methods

2-1. Authors should specify how many cells they used in each experiment.

Response 2-1: We presented number of cells that we used in the text as follows:

- page 2, line 75 and line 90

- page 3, line 108, line138 and line139-140

- page 4, line 169

2-2. Western blot analysis should be written extensively. What they used to run gels? how they transfer the gels, in which kind of membrane? Which secondary antibody they used? How they developed membranes?

Response 2-2: As you pointed out, we added more detailed descriptions for western blot analysis including gel percentage, membrane type, secondary antibody and method of membrane development (page 3, line 130-137, page 4, 143-147).

2-3. Also, in the description of quantitative Real-Time PCR some details are missing, for example it is not clear which method they used to analyze data: fold change? Relative expression procedure?

Response 2-3: We added detailed descriptions in Materials and Method section 2.7 Quantitative Real-Time PCR. We described reaction conditions of real-time PCR in page 4, line 152-155, and also provided analysis method for data in page 4, line 156-157 “Gene expression values were calculated with the 2^ -ΔΔ Ct method. Relative quantification of gene expression was determined by comparison of fold value.”

2-4. In the description of luciferase assay they should provide primers used to amplify promoter regions of human NK2GD and NKp46. How many HEK293 have been seeded and used?

Response 2-4:

- We provided primer sequences for luciferase assay in Materials and Method section as follows in page 4, line 164-167.

“NKp46 primer sequence were forward 5-TTCTCTATCGATAGGTTGGGACTACAGGCATGTGC-3, reverse 5-CCGGAATGCCAAGCTCGCTCAGATTCTGCCGGC-3 and

NKG2D primer sequence were 5-TTTCTCTATCGATAGGGTCAATGGGTACAAAGT-3, 5-CCGGAATGCCAAGCTAATAATGTAAAGATTTAAAAATAGT-3 “

- Also we provided cell number of HEK 293 that we used for luciferase assay in Material and Method section (page 4, line 169)

2-5. Which analysis tool have been used to performed statistical analysis?

Response 2-5: We performed statistical analysis by t-test using GraphPad prism software and Microsoft Excel and we described it in page 4, line 178-179.

3. Results. As I said above, results are mainly descriptive. Authors should perform quantification of all the data they present, and they should show all p values throughout the text. Please perform statistical analysis for all the data (Figure 2A, 2B, 3A, 3B, 3C, 4A, 4B, 4C, 4D, 4E, 5A, 5B, 5C, 6A).

: As you suggest we performed statistical analysis for all the data including Figure 2A and B (page7), 3A, B and C (page 9), 4A, B, C, D and E (page 10), 5A, B and C (page 11) and 6A (page13).

3-1. Page 4, line 147 “thyroid cancer cell unit inhibit NK cell cytolytic function and NK receptor expression” These results are really similar to the results published by the same authors in Front Immunol. 2018; 9: 1859, and I don’t think these results add much more.

Response 3-1: There are differences in experimental methods between the two papers. In previous paper (Front Immunol. 2018; 9: 1859), NK cells were cultured in thyroid cancer cell supernatant to identify soluble factors produced from thyroid cancer cells that suppress NK cell functions. In this paper, however, NK cells were co cultured with thyroid cancer cells to investigate NK cell dysfunction caused by direct interaction between NK cells and thyroid cancer cells. In previous study, we reported that NK cytotoxicity and NK receptor expressions were decreased by PGE2 in supernatant of thyroid cancer cells. These results were due to the decrease of ERK and p65 signal in NK cells. In this study, we reported that IDO also decreased the cytotoxicity and receptor expression of NK cells coculture with thyroid cancer cells. When co-cultured with thyroid cancer, IDO decreased the cytotoxicity and receptor expression of NK cells through down regulating STATs signal. Although the results of the decrease in NK cell functions are the similar, it is considered that the meaning of results may be different because of different experimental methods.

3-2. Page 4, lines 150: authors said that in co-culture system with cancer cells, NK cell function was decreased. Here authors should explain which thyroid cancer cells have been used (not only in material and methods section) and why they used these specific thyroid cell lines. Moreover, NK cell function decreased but the level depends on the thyroid cancer cell. Could the authors provide a hypothesis for these differences?

Response 3-2:

- The explanation about thyroid cancer cells was not enough. We added more explanation about thyroid cancer cells to the text in page 2, line 61-63.

- In, previous study, we used papillary thyroid cancer (PTC) cell line and anaplastic thyroid cancer (ATC) cell lines to determine differences in progression of thyroid cancer. TPC-1 cells are PTC with good prognosis, and FRO and 850-5C are ATC derived tumors with the worst prognosis in thyroid cancer. NK cell functions were greatly affected by co-culture with ATC cells. However, the difference the levels of IDO expression and kynurenine was not considerable between two types of cells. NK cells co-cultured with TPC-1 by IDO inhibitor were recovered cytolytic activity of 94%, while co-cultured with FRO and 850-5C were recovered 87% and 81%, respectively. Through this, we think that IDO is one of the factors that inhibit NK cells. In particular, we expect more factors to work in cancer cells environments with poor prognosis. For example, we found that thyroid cancer cells release an immune suppression factor called PGE2 as well as IDO in Front Immunol. PGE2 was more released in ATC than in PTC. We anticipate that extra factors would exacerbate the prognosis by existing more at the ATC.

3-3. Page 4 and page 6, there is a reason why authors showed receptor expression data plotted in two different ways? E.g Figure 1C and Figure 2B. I think data plotted as in Figure 1C are clearer, moreover the quantification with p values should be provide.

Response 3-3: In experiments using human primary NK cells, the range of receptor expression level or MFI value varies from person to person. For example, some people have an MFI value of 100 in NKG2D, but others are expressed in 1000. Despite the high deviation, all the results of the repeated experiments showed the same tendency. Thus, we showed representative data to show more explicit data. But, as you point out, we should have provided statistics. Therefore, we added statistics graph as Supplementary figure 2B.

3-4. Page 5, line 169. Authors used inhibitors factor for TGF-b, PGE and IDO, but they don’t explain it in material and method section.

Response 3-4: As you pointed out, we added descriptions of PGE2 inhibitor, NS398 and TGF-b inhibitor, anti-TGF-b and IDO inhibitor, 1-methyl-DL-tryptophan in Material and Method (page 2, line 75-77).

3-5. Furthermore, authors treated TPC-1 and 850-5c cell lines with IDO-inhibitor, but not FRO cell line. They should show data also for FRO cells, both for NK cytolytic activity and NK receptor expression.

Response 3-5: We also performed experiments with FRO cells. The type of FRO is anaplastic thyroid cancer cells same as 850-5c cells, and the experimental the results about cytotoxicity were also similar. So, we presented only 850-5c data. We provided the result of FRO as supplementary figures 2A and 2B.

3-6. Page 5, lines 171 and 174. Authors said that treatment with the inhibitor of IDO restored NK cell function. However, data do not support this affirmation. It is more correct to say “partially restored” (as they said in line 174), but still they could not assume it until they provide a quantification with relative statistical analysis.

Response3-6: As you pointed out, "partially restored" seems to be the correct expression. Accordingly, we revised the text (page5, line 206). In addition, we performed statistical analysis and changed to bar graph (page 7, Figure 2A).

3-7. Page 5, authors should explain why they treated cells with IFN-gamma.

Response 3-7: IFN-g is known as the inducer of IDO. Thyroid cancer cells did not constitutively expressed IDO, but they expressed IDO by treatment with INF- g. We added the explanation of IFN-g used as IDO inducer in Introduction section and related reference #[10] in the text (page2, line46).

3-8. Page 6, Fig. 2B. What “Con” stands for? I think “control” however authors should explain it at least in the figure legend.

Response 3-8: As you pointed out, we replaced ‘CON’ with ‘control’ (Figure2, page 7), and added a description to Figure 2 legend (page 7, 8 line 231 - 233).

3-9. Page 7-8, again, provide quantification for western blot analysis (Figure 3C).

Response 3-9: We added quantification data of western blot analysis represented as Figure 3D in page 9.

3-10. Page 8, in Figure 4A authors showed AhR mRNA expression in untreated and kynurenine-treated cells. Please provide error bars and statistical significance.

Response 3-10: We performed statistical analysis and AHR expression was replaced by new graph with error bars and statistical significance (page 10, Figure 4A).

3-11. Page 8 in Figure 1B lower panel, please specify the concentration of kynurenine used for NK treatment and statistical significance.

Response 3-11: We specified the concentration of kynurenine that we used in the graph (page 10, figure4B). Also, we performed significant analysis but there was no statistical significance.

3-12. Page 8, Figure 4E. Please provide quantification and statistical significance of western blot analysis.

Response 3-12: As you suggested, we provided statistical significance of western blot analysis for Figure 4E (page 10) as supplementary figure 5.

3-13. Page 9, Figure 5. Again, for all panels statistical analysis should be provide.

Response 3-13: We performed statistical analysis and added statistical graph next to Figure 5A, next to Figure 5B and below Figure 5C (page 11)

3-14. Page 10, Figure 6A. In this western blot authors analyzed phosphorylated forms of STAT1 and STAT3 in 3 different cell lines. Please, present quantification of western blot analysis and described these cell lines in the material and methods section.

Response 3-14: We presented the quantification of western blot analysis below each band of Figure 6A and we described the analysis method in Figure 6 legend (page 13). The information of three cell lines was described in Material and Methods section (page 2, line 79-85).

3-15. Page 10, lines 10. Which software authors used to verify the presence of STAT1 and STAT3 binding motifs in the promoter region of NK receptors NKp46 and NKG2D?

Response 3-15: We estimated the STAT motif in NK receptor promoters using FIMO from the MEME suite (http://meme-suite.org/tools/fimo) and added this information to the text (page12, line 327).

3-16. Page 10, the results of this luciferase assay are interesting. Nevertheless, authors should perform this experiment on a thyroid cell line

Response 3-16: We tried to examine the relationship between activated STATs and NK receptor expression that were reduced by kynurenine in NK cells. It would be better if we performed experiment with NK cells to observe the correlation between NK receptors and STATs. However, as we showed in Figure 3 and 4, NK cells are unsuitable to confirm the function of activity of STAT because they were already activated. Furthermore, it is experimentally difficult to transfect vectors into NK cells so we examined the interaction between STAT and NK receptors using 293T cells which are suitable to transfection experiments.

Reviewer 2 Report

The study of Park  et al. is trying to uncover the molecular mechanisms that control the expression of IDO in cancer cells and the immune function inhibitory effect of kynurenine metabolite when NK cells interact with cancer cells. They implement a co-culture system of NK cells and thyroid cancer cells (cell lines) in which they measure NK cell cytotoxicity, NK receptor expression, and production of IDO and kynurenine in different conditions (upon inhibition of IDO, inhibition of STATs that they claim to mediate NK cell activity, and inhibition of the aryl hydrocarbon receptor (AhR) in NKs, which serves a a receptor for kynurenine).  

They show that NK cytotoxicity is reduced when human NKs are co-cultured with the cell lines and that IDO expression is induced catabolyzing tryptophan (trp) to Kynurenine. They also claim that NK receptor expression is down-regulated as a result, an effect that is rescued when they use an IDO inhibitor. They also show that STAT1 and STAT3 phosphorylated levels are decreased in the presence of Kynurenine while total protein levels are stable. As it has been previously described the AhR receptor serves as a kynurenine receptor in NKs, so when they use an AhR antagonist they reverse the effect of kynurenine in cytotoxicity, expression levels of NK receptors and STAT signalling (STAT1, STAT3). Further they use a luciferase assay to show that STAT1 and STAT3 promote the expression of NKG2D and NKp40 receptors by binding to regulatory elements in their proximal promoter.

However, there are some points that should be addressed further in order to support overall statements throughout the article. In particular:

In Fig.1C the % of receptor  expression is presented but it is not clear if this is MFI or % of cells positive for these receptors. In line 152 "number of NK cell receptors" should be changed.

In Fig2B the FACS plots (with MFI) are representative and statistics are not present to support the statements in the text (line176).

In Fig2C it should be explained why in the 850-5C cell line (condition +NK) the expression of IDO is lower than (+INFg) alone, while in the other two cell lines is higher in that condition.

In Fig 2D it is shown that co-culture of NKs with the cell lines induces INFg expression, avg 4 times more than NK only). If INFg induces expression of IDO then why in 850-5C cells (highest INFg levels) there is less IDO and kynurenine conc is induced similarly to the other two cell lines (fig 2F). Please explain...

In Fig3B again MFI stats are missing..., similar to Fig 4A and Fig4B &Fig4D.

In Fig 5B the compound effect of the inhibitors of STAT1 and STAT3 seems to affect more the cytoxicity of NKs to K562 cells as compared to kynurenine alone. Is that something to be expected? Please explain...

In Fig 5C stats are missing form the fact plots.

Additionally it is not clear why the authors have chosen to study the effect of STAT1 and STAT3 only when it is known that STAT4 and Nfkb bind to INFg promoter, which is important for the induction of IDO.

In Fig6A the protein gel is not equally  loaded and so the statement in line 285-286 is not strongly supported.

Regarding the STAT motifs (line 283), the authors should precisely indicate the coordinates in the genome and upstream of the NK receptor genes that are located these motifs.

Finally, the discussion it should be more elaborated in interpreting the findings and novelties of the study instead of repeating the results.

Author Response

Response to reviewer 2 comments

1. In Fig.1C the % of receptor expression is presented but it is not clear if this is MFI or % of cells positive for these receptors. In line 152 "number of NK cell receptors" should be changed.

Response 1: The ‘% of receptor expression’ that we used means the percentage of positive cells expressing NK receptor. We modified the sentence "number of NK cell receptors" to” percentage of positive cells expressing NK cell receptors" (page 5, line 186).

2. In Fig2B the FACS plots (with MFI) are representative and statistics are not present to support the statements in the text (line176).

Response 2: To support the statements of line 211(lines changed from 176), we provided statistical data of MFI values for Figure 2B (page 7) in Supplementary figure 2.

3. In Fig2C it should be explained why in the 850-5C cell line (condition +NK) the expression of IDO is lower than (+INF g) alone, while in the other two cell lines is higher in that condition.

Response 3: We co-cultured three different thyroid cancer cells with NK cells, respectively. When NK cells were co-cultured with three different thyroid cancer cells, respectively, the amount of IFN-g released from NK cells was similar. However, IDO expression level from 850-5C was lower than those of the other two cells. There are two possibilities for these results. The first, other studies suggest that IDO is induced independently of IFN-g indicating that IDO can have different characteristics depending on the tissue or cells (Adv Exp Med Biol. 1999; 467:553-7.). Second, IDO may be the result of a mechanism regulated by other factors except INF-g under co-cultured conditions with NK cells. But we have not yet figured out exactly what the cause is. Therefore, further research is needed. In particular, we will clarify additional factors related to the mechanism of IDO.

4. In Fig 2D it is shown that co-culture of NKs with the cell lines induces INF g expression, avg 4 times more than NK only). If INF- g induces expression of IDO then why in 850-5C cells (highest INF g levels) there is less IDO and kynurenine conc is induced similarly to the other two cell lines (fig 2F). Please explain...

Response 4: IDO is mainly known to be induced by IFN- g, and our results also confirm that the level of IDO increased when IFN- g was treated. In contrast, expression of IDO was lower in 850-5c than in the other two thyroid cell lines even though co-cultured NK cells released similar level of IFN- g. For this reason, we guess that IDO is regulated to other factors besides IFN-r after stimulated by IFN- g. For other reasons, we guess that TPC-1 and FRO cells can regulate the level of IDO using kynurenine which regulates the level of IDO. To confirm this mechanism further study will be needed.

5. In Fig3B again MFI stats are missing..., similar to Fig 4A and Fig4B &Fig4D.

Response 5: We added all of statistics data that you pointed out including Figure3B (page 9), Figure 4A, B and D (page 10)

6. In Fig 5B the compound effect of the inhibitors of STAT1 and STAT3 seems to affect more the cytotoxicity of NKs to K562 cells as compared to kynurenine alone. Is that something to be expected? Please explain...

Response 6: The STAT family, including STAT1 and STAT3, is mainly activated by cytokine stimulation. We thought that STAT is activated because NK cells are cultured in medium containing cytokines. STAT1 and STAT3 are also known to be involved in NK cytotoxicity and cytokine secretion. We expected that inhibitors of STAT1 and STAT3 would cause decrease of NK cell activity. However, the inhibitors of STATs more strongly decreased NK activity than kynurenine. The activated STATs are thought to be more potent because they are directly related to the regulation of NK cell functional activity as transcription factors. Thus, we assume that kynurenine may be upstream of the STAT signal.

7. In Fig 5C stats are missing form the fact plots.

Response 7: As you suggest, we added statistical graph below Figure 5C (page 11).

8. Additionally, it is not clear why the authors have chosen to study the effect of STAT1 and STAT3 only when it is known that STAT4 and Nfkb bind to INFg promoter, which is important for the induction of IDO.

Response 8: We agree with your opinion. We already investigated signals including STAT4 and STAT5 which were reduced by kynurenine in NK cells, and the level of STAT4 and STAT5 were unchanged. We added those results in Supplementary figure 4 and this explanation to Discussion section (page 14, line 381-385). The level of NF-kb (detection as p65) was also unchanged which was already represented in Supplementary figure 4. Only the activation of STAT1 and STAT3 were reduced by kynurenine in NK cells. Thus, we focused study on STAT1 and STAT3 signaling.

9. In Fig6A the protein gel is not equally loaded and so the statement in line 285-286 is not strongly supported.

Response 9: We recognized that the loading amount was not equal. Therefore, we normalized the intensity of the target band based on actin intensity. We presented the quantitative results below each band. Nevertheless, we can confirm that the expression of activated STATs in 293T is low and this support statement in line 331-333 (lines changed from 285-286).

10. Regarding the STAT motifs (line 283), the authors should precisely indicate the coordinates in the genome and upstream of the NK receptor genes that are located these motifs

Response 10: We estimated the stat motif in NK receptor promoters using FIMO from the MEME suite (http://meme-suite.org/tools/fimo). We also added information to the text (page12, line326-327). We provided the information of NK promoter sequences and represented STAT motif in Supplementary figure 6.

11. Finally, the discussion it should be more elaborated in interpreting the findings and novelties of the study instead of repeating the results.

Response 11: As you suggest, we added statements to Discussion section in page 14, line 380-385.

Reviewer 3 Report

In the present manuscript, Park et al. investigated the ability of Indoleamine-2,3-Dioxygenasi (IDO) of thyroid cancer cells to suppress Natural Killer (NK) cells activity through STAT-mediated down-regulation of NKp46 and NKG2D expression. The authors found that kynurenine, an IDO metabolite produced by thyroid cancer cells, was able to enter NK cells through the aryl hydrocarbon receptor (AhR) and to inhibit the expression of different NK cells receptors, mostly NKp46 and NKG2D. In addition, the authors reported that the down-regulated receptor expression depended on the reduced activity of STAT3 and STAT1 transcription factors that controlled NKp46 and NKG2D expression.

The manuscript is interesting, but some issues should be addressed by the authors to make the manuscript suitable for publication on JCM.

1.      In figures 1A and 1B the authors showed that NK cells vitality was slightly affected by TPC-1 co-culture respect to NK cells only, while it decreased about 50% following 8505C or FRO cells co-culture. In figure 2A the extent of NK cells death in the presence of TPC-1 cells was much higher (about 50%) than that of figures 1A and 1B and comparable to that of 8505C in figures 1A, 1B and 2A. The authors should explain the discrepancy of TPC-1 data in the two figures.

2.      NK cells receptor expression was measured as percentage of expression in figure 1C and as mean fluorescence intensity in figure 2B. In the presence of TPC-1 or 8505C cells, the percentage of NKp44 expression was unaffected respect to the control (fig. 1C), while the mean fluorescence intensity decreased about 50% (fig. 2B). By the same way, in figure 2B the authors reported a 5-fold down-regulation of NKp46 mean fluorescence intensity  respect to control, while only a 1.5-fold decrease of NKp46 expression was described in figure 1C. Also in this case data from figures do not match. This is an important issue because it should be demonstrated the extent of receptor expression restoration after IDO inhibition in TPC-1 cells on the basis of results of figure 2A or figure 1A.

3.      The authors chose to over-express STAT1 and STAT3 in 293T cells to activate exogenous NKp46 or NKG2D promoters thereby demonstrating STAT1- and STAT3-mediated transcriptional regulation of both receptor expression (fig. 6B). Given that they showed a strong basal activity of both STAT1 and STAT3 in NKL cells (fig. 6A), it should be more appropriated to inhibit their expression in NKL cells, a more physiological experimental system than 293T cells, by RNA silencing and to evaluate endogenous NKp46 and NKG2D expression following STAT1 and STAT3 down-regulation. In addition, the authors could be improve results of figure 6 by chromatin immunoprecipitation analysis of NKp46 and NKG2D promoters in the presence and in the absence of STAT1 and STAT3 in NKL cells.

Author Response

Response to reviewer 3 comments

1. In figures 1A and 1B the authors showed that NK cells vitality was slightly affected by TPC-1 co-culture respect to NK cells only, while it decreased about 50% following 8505C or FRO cells co-culture. In figure 2A the extent of NK cells death in the presence of TPC-1 cells was much higher (about 50%) than that of figures 1A and 1B and comparable to that of 8505C in figures 1A, 1B and 2A. The authors should explain the discrepancy of TPC-1 data in the two figures.

Response 1: In experiments using human primary NK cells, the variations of experiments were high but the tendency was similar. There was a large deviation in mean cytotoxicity value of NK cells cocultured with TPC-1 cells between original Figure 1 and original Figure 2. We added two additional experiments to increase number of samples on Figure 2A and the deviation between the averages of two graphs was reduced. Thus, we replaced the new Figure 2A (page 7) with the original Figure 2A and provided a statistic of the data.

2. NK cells receptor expression was measured as percentage of expression in figure 1C and as mean fluorescence intensity in figure 2B. In the presence of TPC-1 or 8505C cells, the percentage of NKp44 expression was unaffected respect to the control (fig. 1C), while the mean fluorescence intensity decreased about 50% (fig. 2B). By the same way, in figure 2B the authors reported a 5-fold down-regulation of NKp46 mean fluorescence intensity respect to control, while only a 1.5-fold decrease of NKp46 expression was described in figure 1C. Also, in this case data from figures do not match. This is an important issue because it should be demonstrated the extent of receptor expression restoration after IDO inhibition in TPC-1 cells on the basis of results of figure 2A or figure 1A.

Response 2: In the case of MFI, the range of receptor expression varied from experiment to experiment. For example, some samples have an MFI value of 100 in NKG2D but others have an MFI value of 1000. Therefore, we used the percentage of positive cells expressing NK cell receptors to represent in same graph as in Figure 1C. However the extent of the change in NK receptor expression tends to be reduced by representing the percentage of NK cell receptor expression. This maybe because the expression of some receptors is not clearly separated between positive and negative parts in graph. Thus, we showed representative data to show clear results as Figure 2B and we added statistical data for Figure 2B in Supplementary figure 2B to increase the reliability.

3. The authors chose to over-express STAT1 and STAT3 in 293T cells to activate exogenous NKp46 or NKG2D promoters thereby demonstrating STAT1- and STAT3-mediated transcriptional regulation of both receptor expression (fig. 6B). Given that they showed a strong basal activity of both STAT1 and STAT3 in NKL cells (fig. 6A), it should be more appropriated to inhibit their expression in NKL cells, a more physiological experimental system than 293T cells, by RNA silencing and to evaluate endogenous NKp46 and NKG2D expression following STAT1 and STAT3 down-regulation. In addition, the authors could be improve results of figure 6 by chromatin immunoprecipitation analysis of NKp46 and NKG2D promoters in the presence and in the absence of STAT1 and STAT3 in NKL cells.

Response 3: In general, RNA silencing-related factors are transfected well, but it is difficult to transfect the DNA vector in NK cells. We also tried transfection experiment using vector containing NK receptor promoter on primary NK cells as well as NK92 cell line, but the transfections did not work well. Thus, we chose 293T cells that are more susceptible to transfection.

Round 2

Reviewer 3 Report

The authors addressed all the issues raised by the referee..